# Probabilistic forecasting of replication studies

**Samuel Pawel**[¤]*, **Leonhard Held**[¤]

Epidemiology, Biostatistics and Prevention Institute (EBPI), Center for Reproducible Science (CRS), University of Zurich, Zurich, Switzerland

¤ Current address: Epidemiology, Biostatistics and Prevention Institute, Zurich, Switzerland
* samuel.pawel@uzh.ch

## Abstract

Throughout the last decade, the so-called replication crisis has stimulated many researchers to conduct large-scale replication projects. With data from four of these projects, we computed probabilistic forecasts of the replication outcomes, which we then evaluated regarding discrimination, calibration and sharpness. A novel model, which can take into account both inflation and heterogeneity of effects, was used and predicted the effect estimate of the replication study with good performance in two of the four data sets. In the other two data sets, predictive performance was still substantially improved compared to the naive model which does not consider inflation and heterogeneity of effects. The results suggest that many of the estimates from the original studies were inflated, possibly caused by publication bias or questionable research practices, and also that some degree of heterogeneity between original and replication effects should be expected. Moreover, the results indicate that the use of statistical significance as the only criterion for replication success may be questionable, since from a predictive viewpoint, non-significant replication results are often compatible with significant results from the original study. The developed statistical methods as well as the data sets are available in the R package `ReplicationSuccess`.

## Introduction

Direct replication of past studies is an essential tool in the modern scientific process for assessing the credibility of scientific discoveries. Over the course of the last decade, however, concerns regarding the replicability of scientific discoveries have increased dramatically, leading many to conclude that science is in a crisis [1, 2]. For this reason, researchers in different fields, *e. g.* psychology or economics, have joined forces to conduct large-scale replication projects. In such a replication project, representative original studies are carefully selected and then direct replication studies of these original studies are carried out.

By now, many of the initial projects have been completed and their data made available to the public [3–9]. The low rate of replication success in some of these projects has received enormous attention in the media and scientific communities. Moreover, these results lead to an increased awareness of the replication crisis as well as to increased interest in research on the scientific process itself (*meta-science*).

**Data Availability Statement:** All relevant data are within the paper, its Supporting Information files, and the osf repository https://osf.io/gqfv7/.

**Funding:** The preparation of this article was partially supported by the Swiss National Science Foundation (project number 189295 awarded to

LH). The funders had no role in study design, data collection and analysis, decision to publish, or preparation of the manuscript. There was no additional external funding received for this study.

**Competing interests:** The authors have declared that no competing interests exist.

Making forecasts about an uncertain future is a common human desire and central for decision making in science and society [10, 11]. There have been many attempts to forecast the outcomes of replication studies based on the results from the original studies [5, 7, 12–15]. This is interesting for various reasons: First, a forecast of how likely a replication will be "successful" according to some criterion (*e. g.* an effect estimate reaches statistical significance) can help to assess the credibility of the original finding in the first place and inform the decision whether a replication study should be conducted at all. Second, after a replication has been completed, its results can be compared to its forecast in order to assess compatibility between the two findings. Finally, forecasting can also be helpful in designing an informative replication study, for example it can be used for sample size planning.

Although there have been theoretical contributions to the literature long before the replication crisis started [16–18], the last years have witnessed new developments regarding forecasting of replication studies. Moreover, due to the increasing popularity of replication studies, forecasts could be evaluated with actual data.

For instance, *prediction markets* have been used in order to estimate the peer belief about whether a replication will result in a statistically significant outcome [5, 7, 12, 15]. Prediction markets are a tool to aggregate beliefs of market participants regarding the possibility of an investigated outcome and they have been used successfully in numerous domains, *e. g.* sports and politics. However, despite good predictive performance, taking statistical significance as the target variable of the forecasts requires arbitrary dichotomization of the outcomes, although one would prefer to rather forecast the replication effect estimate itself. Moreover, the evaluation of these forecasts was usually based on ad-hoc measures such as correlation of the estimated probabilities with the outcome. In fields where forecasting is of central importance, *e. g.* meteorology, climatology, or infectious disease epidemiology, extensive methodology has been developed to specifically assess calibration, discrimination, and sharpness of probabilistic forecasts [11]. It is therefore of interest to assess whether more insights about the forecasts can be gained when applying a more state-of-the-art evaluation strategy. Finally, it is also of interest to benchmark the prediction market forecasts with statistical forecasts that do not require recruiting experts and setting up prediction market infrastructure.

A statistical method to obtain probabilistic forecasts of replication estimates was proposed by Patil, Peng, and Leek [13] and then also used in the analysis of the outcomes of some large-scale replication projects. Specifically, the agreement between the original and replication study was assessed by a prediction interval of the replication effect estimate based on the original effect estimate. This method was illustrated using the data set from the *Reproducibility Project: Psychology* [4], and it was also used in the analyses of the *Experimental Economics Replication Project* [5] and the *Social Sciences Replication Project* [7]. In all of these analyses, the coverage of the 95% prediction intervals was examined to assess predictive performance. Although this evaluation method provides some clue about the calibration of the forecasts, more sophisticated methods exist to assess calibration and sharpness specifically [11]. Moreover, the prediction model which was used does not take into account that the original effect estimates may be inflated. In the statistical prediction literature, the phenomenon that future observations of a random quantity tend to be less extreme than the original observation, is commonly known as *regression to the mean* and usually addressed by shrinkage methods [19]. This effect might be even more pronounced by the influence of publication bias [20, 21] or questionable research practices [22, 23]. Finally, the model from Patil et al. [13] also makes the naive assumption that the effect estimates from both studies are realizations of the same underlying effect size, however, there is often between-study heterogeneity [24, 25]. This can be caused, for example, by different populations of study participants or different laboratory equipment being used in the original and replication study.

The objective of this paper is to improve on the previous statistical forecasts and also on their evaluation. In particular, we will develop and evaluate a novel prediction model which can take into account inflation of the original effect estimates as well as between-study heterogeneity of effects. With the available data from large-scale replication projects, we aim to predict the effect estimates of the replication studies based on the estimates from the original studies and knowledge of the sample size in both studies. We will assess the forecasts regarding discrimination, calibration, and sharpness using state-of-the-art evaluation methods from the statistical prediction literature. Finally, we will also benchmark them with the forecasts from prediction markets and the naive model which was used so far.

It is worth pointing out that in our forecasting approach the link between original and replication study is based only on information from the original study and the sample size of the replication study. This is fundamentally different from approaches where the link between original and replication study is estimated from a training sample of past original and replication study pairs, as for example done recently by Altmejd et al. [14]. Since in our approach no replication estimates are used to estimate any parameter, all evaluations presented in this paper provide "out-of-sample" performance measures, thereby eliminating the need to split the data and perform cross-validation.

The structure of this paper is as follows: descriptive results on the data collected are presented in the following section. We then develop a novel model of effect sizes that addresses the shortcomings of the model used in previous analyses. Next, we compute forecasts for the collected data and systematically evaluate and compare them with forecasts based on the previously used model. Finally, the paper ends with a discussion of the results obtained.

## Data

Data from all replication projects with a "one-to-one" design (*i. e.* one replication for one original study) that are, to our knowledge currently available, were collected. The R code and details on data preprocessing can be found in S1 Appendix. In all data sets, effect estimates were provided as correlation coefficients (*r*). An advantage of correlation coefficients is that they are bounded to the interval between minus one and one and are thus easy to compare and interpret. Moreover, by applying the variance stabilizing transformation, also known as Fisher *z*-transformation, $\hat{\theta} = \tanh^{-1}(r)$, the transformed correlation coefficients become asymptotically normally distributed with their variance only being a function of the study sample size *n*, *i. e.* $\mathrm{Var}(\hat{\theta}) = 1/(n-3)$ [26].

### Reproducibility project: Psychology

In the *Reproducibility Project: Psychology*, 100 replications of studies from the field of psychology were conducted [4]. The original studies were published in three major psychology journals in the year 2008. Only the study pairs of the "meta-analytic subset" were used, which consists of 73 studies where the standard error of the Fisher *z*-transformed effect estimates can be computed [27].

### Experimental economics replication project

This project attempted to replicate 18 experimental economics studies published between 2011 and 2015 in two high impact economics journals [5]. For this project a *prediction market* was also conducted in order to estimate the peer beliefs about whether a replication will result in a statistically significant result. Since the estimated beliefs are also probabilistic predictions, they

can be compared to the probability of a significant replication effect estimate under the statistical prediction models.

## Social sciences replication project

This project involved 21 replications of studies on the social sciences published in the journals *Nature* and *Science* between 2010 and 2015 [7]. As in the experimental economics replication project, a prediction market to estimate peer beliefs about the replicability of the original studies was conducted and the resulting belief estimates can be used as a comparison to the statistical predictions.

## Experimental philosophy replicability project

In this project, 40 replications of experimental philosophy studies were carried out. The original studies had to be published between 2003 and 2015 in one of 35 journals in which experimental philosophy research is usually published (a list defined by the coordinators of this project) and they had to be listed on the experimental philosophy page of the Yale university [8]. Effect estimates on correlation scale and effective sample size for both the original and replication were only available for 31 study pairs. Our analysis uses only this subset.

## Descriptive results

Fig 1 shows plots of the original versus the replication effect estimate, both on the correlation scale. Most effect estimates of the replication studies are considerably smaller than those of the original studies. In particular, the mean effect estimates of the replications are roughly half as large as the mean effect estimates of the original studies. This is not the case for the philosophy project, however, where the mean effect estimate only decreased from 0.39 to 0.34. Furthermore, studies showing a comparable effect estimate in the replication and original study usually also achieved statistical significance, while studies showing a large decrease in the effect estimate were less likely to achieve statistical significance in the replication.

Fig 2 illustrates the elicited prediction market beliefs about whether the replication studies will achieve statistical significance. In the case of the economics data set, the distribution of the prediction market beliefs is very similar for significant and non-significant replications. In the social sciences project, on the other hand, the elicited beliefs separate significant and non-significant replications completely for a cut-off around 0.55.

## Methods

To introduce some notation, denote the overall effect by $\theta$, study-specific underlying effects by $\theta_o$ and $\theta_r$, and their estimates by $\hat{\theta}_o$ and $\hat{\theta}_r$, with the subscript indicating whether they come from the original or the replication study. Let the corresponding standard errors be denoted by $\sigma_o$ and $\sigma_r$ and also let the heterogeneity variance be $\tau^2$. Similarly, define the variance ratio as $c = \sigma_o^2/\sigma_r^2$, the relative between-study heterogeneity as $d = \tau^2/\sigma_o^2$, and also denote the corresponding test statistics by $t_o = \hat{\theta}_o/\sigma_o$ and $t_r = \hat{\theta}_r/\sigma_r$. Finally, let $\Phi(x)$ be the cumulative distribution function of the standard normal distribution evaluated at $x$ and let $z_\alpha$ denote the $1 - \alpha$ quantile thereof.

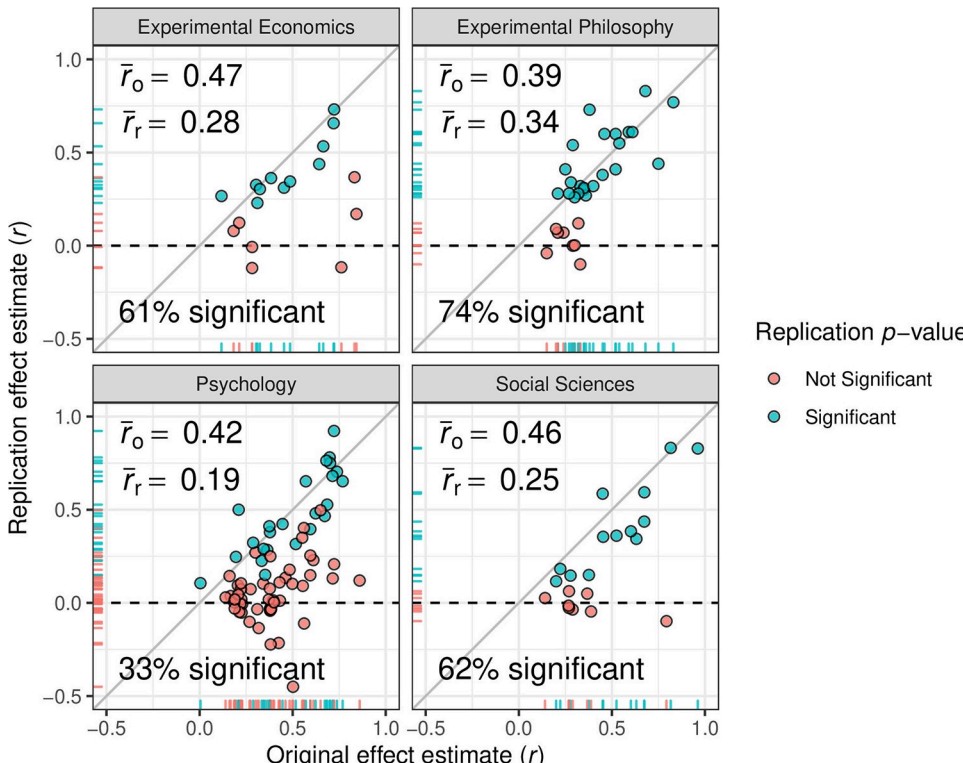

**Fig 1. Original effect estimate versus replication effect estimate.** Effect estimates are on correlation scale. The color indicates whether statistical significance at the (two-sided) 0.05 level was achieved.

We propose the following Bayesian hierarchical model for the effect estimates

$$\hat{\theta}_k \,|\, \theta_k \sim \mathrm{N}(\theta_k, \sigma_k^2) \tag{1a}$$

$$\theta_k \,|\, \theta \sim \mathrm{N}(\theta, \tau^2) \tag{1b}$$

$$\theta \sim \mathrm{N}(\mu_\theta, \sigma_\theta^2) \tag{1c}$$

where $\sigma_k^2, \tau^2, \mu_\theta, \sigma_\theta^2$ are fixed and $k \in \{o, r\}$ (see Fig 3 for a graphical illustration). After a suitable transformation a large variety of effect size measures are covered by this framework (*e. g.* mean differences, odds ratios, correlations). For instance, $\hat{\theta}_k = \tanh^{-1}(r_k)$ and

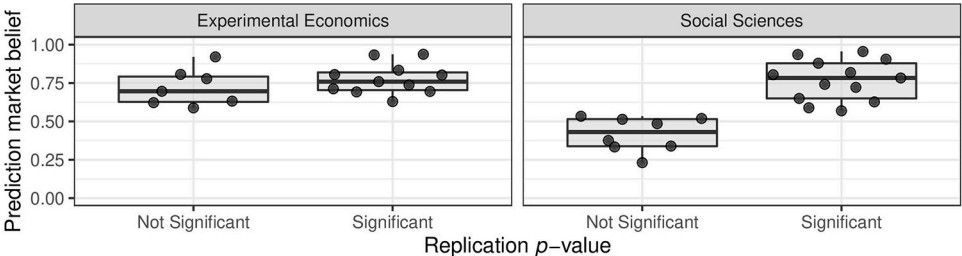

**Fig 2. Statistical significance of replication effect estimate versus prediction market belief.** The significance threshold $\alpha = 0.05$ (two-sided) is used in all cases.

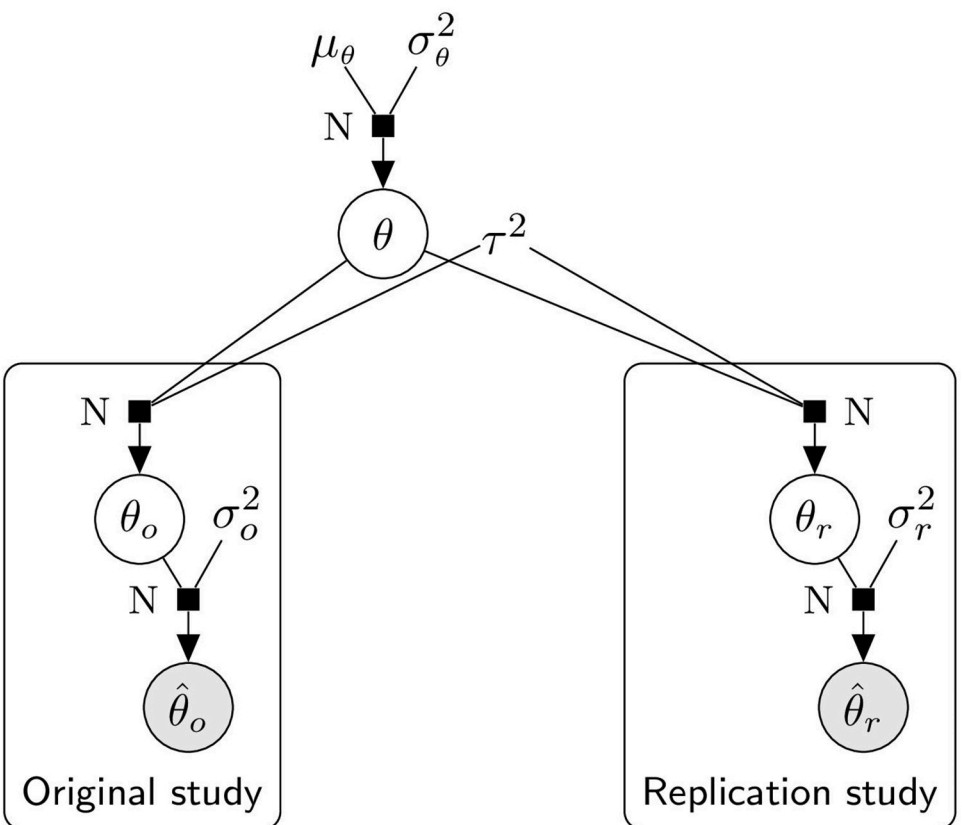

**Fig 3. Hierarchical model of effect sizes in replication setting.** Random variables are encircled (and grey if they are observable).

$\sigma_k^2 = 1/(n_k - 3)$ are used in the four data sets for our analysis. The normality assumption is also common to many meta-analysis methods. Together with a fixed heterogeneity variance $\tau^2$, it leads to analytical tractability of the predictive distributions. In this model, the case where effect estimates of the original and replication studies are not realizations of the same, but of slightly different underlying random variables is taken into account and controlled by the heterogeneity variance $\tau^2$. That is, for the limiting case $\tau^2 \to 0$, the study-specific underlying effects $\theta_o$ and $\theta_r$ are assumed to be the same, while for the other extreme $\tau^2 \to \infty$, $\theta_o$ and $\theta_r$ are assumed to be completely unrelated. Furthermore, the choice of the prior distribution of $\theta$ provides additional flexibility to incorporate prior knowledge about the overall effect. In the following, the predictive distributions of the replication effect estimate under two interesting prior distributions are discussed.

### Flat prior

If the prior distribution Eq (1c) is chosen to be flat, the posterior distribution of the overall effect $\theta$ after observing the original study effect estimate becomes $\theta \,|\, \hat{\theta}_o \sim \mathrm{N}(\hat{\theta}_o, \sigma_o^2 + \tau^2)$. The posterior predictive distribution of $\hat{\theta}_r$ then turns out to be

$$\hat{\theta}_r \,|\, \hat{\theta}_o \sim \mathrm{N}(\hat{\theta}_o, \sigma_o^2 + \sigma_r^2 + 2\tau^2). \tag{2}$$

Under this predictive model, one implicitly assumes the effect estimation in the original study to be unbiased, since the predictive density is centered around the original effect estimate.

Furthermore, the uncertainty coming from the original and replication study, as well as the uncertainty from the between-study heterogeneity is taken into account. Also note that for $\tau^2 = 0$, Eq (2) reduces to the naive model from Patil et al. [13] used in previous analyses.

Given this predictive model, the test statistic of the replication is distributed as

$$t_r \,|\, t_o \sim \mathrm{N}(\sqrt{c} \cdot t_o, c + 1 + 2cd), \tag{3}$$

which only depends on the original test statistic $t_o$, the variance ratio $c$, and the relative heterogeneity $d$. From Eq (3) one can easily compute the power to obtain a statistically significant result in the replication study. Hence, this generalizes the *replication probability* [16, 17], *i. e.* the probability of obtaining a statistically significant finding in the same direction as in the original study, to the setting of possible between-study heterogeneity.

## Sceptical prior

Instead of a flat prior, one can also choose a normal prior centered around zero for Eq (1c), reflecting a more sceptical belief about the overall effect [28]. Moreover, we decided to use a parametrization of the variance parameter inspired by the *g*-prior [29] known from the regression literature, *i. e.* $\theta \sim \mathrm{N}(0, g \cdot [\sigma_o^2 + \tau^2])$ with fixed $g > 0$. A well-founded approach to specify the parameter $g$ when no prior knowledge is available is to choose it such that the marginal likelihood is maximized (empirical Bayes estimation). In doing so, the empirical Bayes estimate $\hat{g} = \max\{\hat{\theta}_o^2/(\sigma_o^2 + \tau^2) - 1, 0\}$ is obtained. Fixing $g$ to $\hat{g}$ and applying Bayes' theorem, the posterior distribution of the overall effect $\theta$ after observing the original effect estimate becomes $\theta \,|\, \hat{\theta}_o, \hat{g} \sim \mathrm{N}(s \cdot \hat{\theta}_o, s \cdot [\sigma_o^2 + \tau^2])$, with shrinkage factor

$$s = \frac{\hat{g}}{1 + \hat{g}} = \max\left\{1 - \frac{1+d}{t_o^2}, 0\right\}. \tag{4}$$

Fig 4 shows the shrinkage factor $s$ as a function of the relative between-study heterogeneity $d$ and the test statistic (or the two-sided $p$-value) of the original study. Interestingly, for $d = 0$, Eq (4) reduces to the factor known from the theory of optimal shrinkage of regression coefficients [19, 30].

The posterior predictive distribution of $\hat{\theta}_r$ under this model becomes

$$\hat{\theta}_r \,|\, \hat{\theta}_o \sim \mathrm{N}(s \cdot \hat{\theta}_o, s \cdot (\sigma_o^2 + \tau^2) + \sigma_r^2 + \tau^2). \tag{5}$$

The promise of this method is that shrinkage towards zero should improve predictive

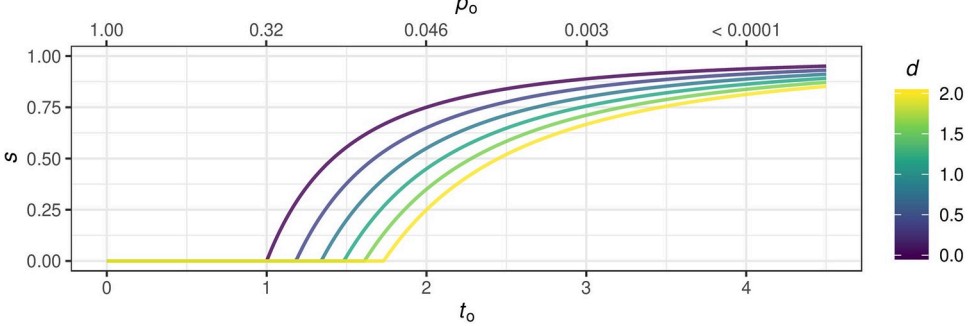

**Fig 4. Evidence-based shrinkage.** Shrinkage factor $s$ as function of the test statistic $t_o$ (bottom axis) and the two-sided $p$-value $p_o$ (top axis) of the original study and the relative between-study heterogeneity $d = \tau^2/\sigma_o^2$.

performance by counteracting the regression to the mean effect. As such, shrinkage also counteracts effect estimate inflation caused by publication bias to some extent. That is, the contribution of the original effect estimate to the predictive distribution shrinks depending on the amount of evidence in the original study (*evidence-based shrinkage*). The less convincing the result from the original study, *i. e.* the smaller $t_o$, the more shrinkage towards zero. On the other hand, shrinkage decreases for increasing evidence and in the limiting case the predictive distribution is the same as under the flat prior, *i. e.* $s \rightarrow 1$ for $t_o \rightarrow \infty$. Moreover, the shrinkage factor in Eq (4) is also influenced by the ratio $d/t_o^2$. If the test statistic is not substantially larger than the relative between-study heterogeneity, *i. e.* $t_o^2 \gg d$, heterogeneity also induces shrinkage towards zero.

Based on this predictive model, the distribution of the test statistic of the replication study depends only on the relative quantities $c$, $d$, and $t_o$. From Eq (6), it is again straightforward to compute the power for a significant replication outcome.

$$t_r \,|\, t_o \sim \mathrm{N}(s \cdot \sqrt{c} \cdot t_o, s \cdot (c + cd) + 1 + cd), \tag{6}$$

Fig 5 shows the replication probability as a function of the original test statistic $t_o$ (or two-sided $p$-value $p_o$) and for different values of the variance ratio $c$. Note that the curves of the shrinkage methods stay constant until $t_o$ reaches a point where Eq (4) starts to become larger than zero. If the original study showed a "just significant result" ($t_o \approx 1.96$) and the precision is equal in the original and the replication study ($c = 1$), the replication probability is just 0.5 when a flat prior is used. This surprising result was already noted two decades ago [16], yet it has not become part of statistical literacy and many practitioners of statistics are still perplexed when they hear about it. If a sceptical prior is used, the replication probability becomes even lower. Moreover, when the precision of the replication is smaller ($c < 1$), the replication probability is also lower, whereas with increased precision ($c > 1$) the replication probability also increases. Finally, for small $t_o$ the replication probability is higher when there is heterogeneity compared to when there is no heterogeneity, while the opposite is true for large $t_o$.

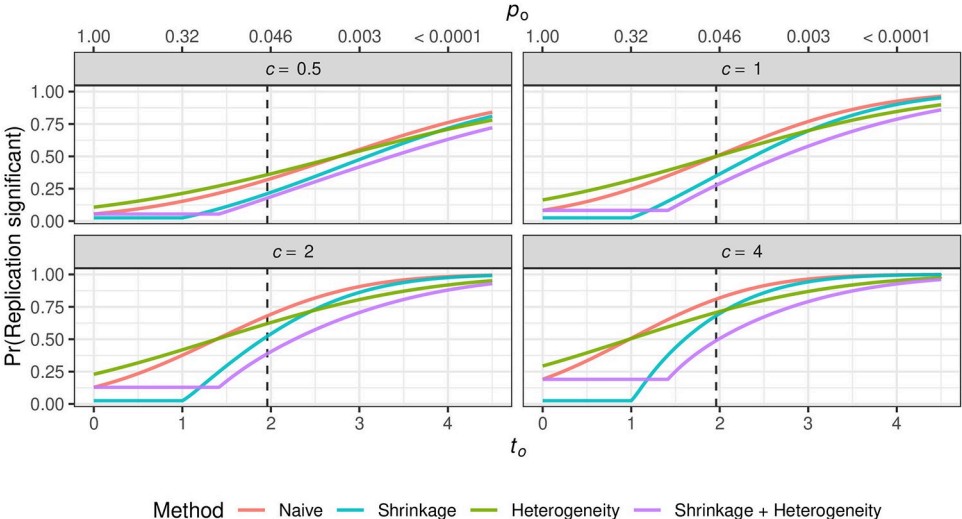

**Fig 5. Replication probability.** Probability of a significant replication outcome in the same direction as in the original study at (two-sided) $\alpha = 0.05$ level as a function of the test statistic $t_o$ (bottom axis) and $p$-value $p_o$ (top axis) of the original study and variance ratio $c = \sigma_o^2/\sigma_r^2$. The dashed line indicates $z_{0.025} \approx 1.96$. In the case of heterogeneity, $d = \tau^2/\sigma_o^2$ is set to one, otherwise to zero.

## Specification of the heterogeneity variance

One needs to specify a value for the heterogeneity variance $\tau^2$ to compute predictions of $\hat{\theta}_r$. However, it is not possible to estimate $\tau^2$ using only the data from the original study, since the overall effect $\theta$ in the marginal likelihood of $\hat{\theta}_o \mid \theta \sim N(\theta, \sigma_o^2 + \tau^2)$ is also unknown.

Ideally, a domain expert would carefully assess original study and replication protocol, and then specify how much heterogeneity can be expected for each study pair individually. This is, however, beyond the scope of this work. We instead want to compare forecasts that use a positive "default value" of $\tau^2$ to forecasts for which $\tau^2$ is set to zero. The goal is then to assess whether or not it makes a difference in predictive performance when heterogeneity is taken into account. Of course we will also investigate how robust our conclusions are to the choice of the default value for $\tau^2$ by conducting a sensitivity analysis (see section "Sensitivity analysis of heterogeneity variance choice").

To determine the value of $\tau^2$, we adapted an approach originally proposed to determine plausible values for $\tau^2$ of heterogeneous log odds ratio effects [28 Chapter 5.7.3, P. 168]: Based on the proposed hierarchical model Eq (1), 95% of the study-specific underlying effects $\theta_k$, $k \in \{o, r\}$ should lie within the interval $\theta \pm z_{0.025} \cdot \tau$. We want to specify a value for $\tau$, such that the range of this interval is not zero, but also not very large, because the whole purpose of a replication study is to replicate an original experiment as closely as possible. The comparison of the limits of this interval is, however, easier if they are transformed to the correlation scale, since $\theta_k$ are a Fisher $z$-transformed correlations $\theta_k = \tanh^{-1}(r_k)$ in all used data sets. We therefore looked at the difference of the transformed 97.5% to the transformed 2.5% quantile, $\delta(\tau) = \tanh(\theta_{k,97.5\%}) - \tanh(\theta_{k,2.5\%})$ as a function of the heterogeneity parameter $\tau$ for an overall effect of $\theta = 0$ (Fig 6). We then chose a value for $\tau$ that lead to a plausible value of $\delta(\tau)$. However, this raises the question of how one should classify these differences and which value should be chosen for the current setting. In the context of power analysis, there exist many classifications of effect size magnitudes, $e.\,g.$ the one by Cohen [31]. We think this classification is appropriate since it was developed to characterize effects in psychology and other social sciences, the fields from which the data at hand are. In this classification a medium effect size should reflect an effect which is "visible to the eye" ($r = 0.3$), a small effect size should be smaller but not trivial ($r = 0.1$), and a large effect size should have the same difference to the medium effect size as the small effect size, but in the other direction ($r = 0.5$). For direct replication studies, we think it is reasonable to assume that the between-study heterogeneity should not be very large, because these kinds of studies are usually matched as closely as possible to the original studies.

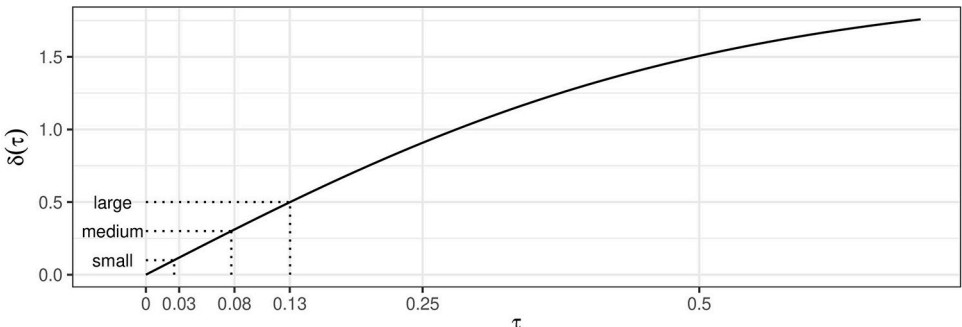

**Fig 6. The difference between backtransformed quantiles $\delta(\tau) = \tanh(\theta_{k,97.5\%}) - \tanh(\theta_{k,2.5\%})$ as a function of between-study heterogeneity $\tau$ for $\theta = 0$.** The values corresponding to small, medium, and large effect sizes on the correlation scale according to the classification by Cohen [31] are depicted by dotted lines.

We therefore chose $\tau = 0.08$, such that $\delta(\tau)$, the difference between the 97.5% and 2.5% quantiles of the study-specific underlying effects, is not larger than the size of a medium effect.

An alternative approach would be to use empirical heterogeneity estimates known from the literature. We therefore also compared the chosen value to the empirical distribution of 497 between-study heterogeneity estimates of meta-analyses with correlation effect sizes in the journal *Psychological Bulletin* between 1990 and 2013 [32] (see S2 Appendix for details). The value of 0.08 corresponds to the 34% quantile of the empirical distribution, which we think is reasonable as those estimates stem from meta-analyses of ordinary studies that are likely to be more heterogeneous than direct replication studies.

## Predictive evaluation methods

A large body of methodology is available to assess the quality of probabilistic forecasts. When comparing the actual observations with their predictive distributions, one can distinguish different aspects. *Discrimination* characterizes how well a model is able to predict different observations. *Calibration*, on the other hand, describes the statistical agreement of the whole predictive distribution with the actual observations, *i. e.* they should be indistinguishable from randomly generated samples from the predictive distribution. One can also assess *sharpness* of the predictions, *i. e.* the concentration of the predictive distribution [11].

Proper scoring rules are an established way to assess calibration and sharpness of probabilistic forecasts simultaneously. We therefore computed the mean logarithmic (LS), quadratic (QS), and continuous ranked probability score (CRPS) for continuous predictive distributions [33], and the mean (normalized) Brier score (BS) for binary predictive distributions [34]. In order to specifically evaluate calibration, several methods were used: First, calibration tests based on scoring rules were conducted, *i. e. Spiegelhalter's z-test* [35] for forecasts with a binary target and four calibration tests based on LS and CRPS [36] for forecasts with a continuous target. All of these tests exploit the fact that under the null hypothesis of perfect calibration, the distribution of certain scores can be determined. Second, the *probability integral transform* (PIT), *i. e.* the value of the predictive cumulative distribution function evaluated at the actual observed value, was computed for each forecast. Under perfect calibration, the PIT values should be uniformly distributed which can be assessed visually, as well as with formal tests [33]. Third, the *calibration slope* method was used to evaluate calibration by regressing the actual observations on their predictions, *i. e.* for forecasts with a binary target using logistic regression. A well calibrated prediction model should lead to a regression slope $\beta \approx 1$, whereas $\beta > 1$ and $\beta < 1$ indicate miscalibration [37]. Finally, to assess the discriminative quality of the forecasts with a binary target, the *area under the curve* (AUC) was computed [38 Chapter 15.2.3].

## Software

All analyses were performed in the R programming language [39]. The full code to reproduce analyses, plots, and tables is provided in S3 Appendix. Methods to compute prediction intervals and to conduct sample size calculations (see S2 Appendix for details), as well as the four data sets are provided in the R package ReplicationSuccess which is available at https://r-forge.r-project.org/projects/replication/.

## Results

In this section, predictive evaluations of four different forecasting methods applied to the data sets are shown: the method with the flat prior and $\tau = 0$, corresponding to the previously used method from Patil et al. [13] (denoted by N for *naive*), the method with the flat prior and

$\tau = 0.08$ (denoted by H for *heterogeneity*), the method with the sceptical prior and $\tau = 0$ (denoted by S for *shrinkage*), and the method with the sceptical prior and $\tau = 0.08$ (denoted by SH for *shrinkage and heterogeneity*).

## Forecasts of effect estimates

In the following, evaluations of forecasts of the replication effect estimates are shown.

**Prediction intervals.** Fig 7 shows plots of the original versus the replication effect estimates. In addition, the corresponding 95% prediction interval is vertically shown around each study pair.

Comparing the different methods across projects, the S method shows similar coverage as the N method in the economics and philosophy data sets (83–84%), whereas in the psychology and social sciences data sets the S method (75–76%) shows a higher coverage compared to the N method (67–70%). As expected, when heterogeneity is taken into account, the prediction intervals become wider and the coverage improves considerably in all cases. In the philosophy and economics projects the highest coverage is achieved for the forecasts from the H method (94%), while in the psychology and social sciences projects the highest coverage is achieved for

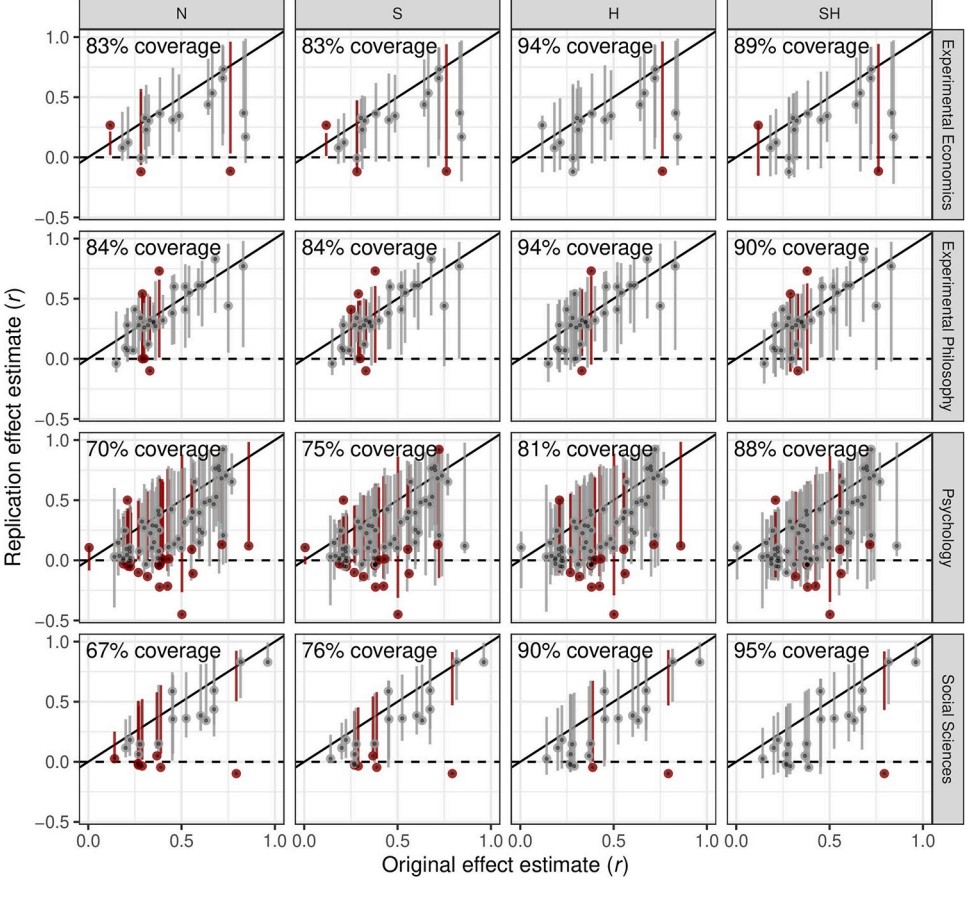

**Fig 7. Original and replication effect estimates with 95% prediction intervals of the replication effect estimates (vertical lines).** Forecasting methods are abbreviated by N for *naive*, S for *shrinkage*, H for *heterogeneity*, SH for *shrinkage and heterogeneity*.

**Table 1. Mean quadratic score (QS), mean logarithmic score (LS), mean continuous ranked probability score (CRPS), and harmonic mean of $p$-values from four score-based calibration tests ($\mathring{p}$).**

| Project | Method | Score Type | | | $\mathring{p}$ |
|---|---|---|---|---|---|
| | | QS | LS | CRPS | |
| Experimental Economics | N | −0.83 | 0.34 | 0.21 | 0.013 |
| $n = 18$ | S | −1.17 | 0.17 | 0.17 | 0.056 |
| | H | −1.14 | 0.18 | 0.21 | 0.24 |
| | SH | −1.32 | 0.02 | 0.17 | 0.79 |
| Experimental Philosophy | N | −1.33 | −0.05 | 0.12 | 0.0005 |
| $n = 31$ | S | −1.46 | −0.06 | 0.12 | 0.0002 |
| | H | −1.51 | −0.18 | 0.12 | 0.81 |
| | SH | −1.67 | −0.20 | 0.11 | 0.66 |
| Psychology | N | −0.07 | 0.87 | 0.22 | < 0.0001 |
| $n = 73$ | S | −0.15 | 0.86 | 0.19 | < 0.0001 |
| | H | −0.55 | 0.51 | 0.22 | < 0.0001 |
| | SH | −0.85 | 0.27 | 0.18 | < 0.0001 |
| Social Sciences | N | −0.17 | 0.85 | 0.22 | < 0.0001 |
| $n = 21$ | S | −0.58 | 0.54 | 0.19 | < 0.0001 |
| | H | −0.67 | 0.55 | 0.21 | < 0.0001 |
| | SH | −1.17 | 0.25 | 0.18 | 0.01 |

Forecasting methods are abbreviated by N for *naive*, S for *shrinkage*, H for *heterogeneity*, SH for *shrinkage and heterogeneity*.

the SH forecasts (88–95%). Moreover, in all but the psychology data set, the best method is able to achieve nominal coverage, whereas in the psychology data set the best method achieves slightly less. These improvements suggest improved calibration of the forecasts which take heterogeneity into account (and shrinkage in the case of the social sciences and psychology data sets). Finally, in the psychology and social sciences projects, the replication effect estimates that are not covered by their prediction intervals tend to be smaller than the lower limits of the intervals. In the economics and philosophy projects, on the other hand, the non-coverage appears to be more symmetric.

**Scores.** Table 1 shows the mean quadratic score (QS), mean logarithmic score (LS), and the mean continuous ranked probability score (CRPS) for each combination of data set and forecasting method. The SH method achieved the lowest mean score for each score type and in all projects, suggesting that this method performs the best among the four methods. The N method, on the other hand, usually showed the highest mean score across all score types, indicating that this method performs worse compared to the other methods. We also tested for deviation from equal predictive performance using paired tests and in most cases there is evidence that the difference between the scores of the SH forecasts and the scores of the other forecasts is substantial (see S2 Appendix for details).

**Calibration tests.** A total of four score-based calibration tests have been performed. These tests exploit the fact that for normal predictions under the null hypothesis of perfect calibration, the first two moments of the distribution of the mean LS and the mean CRPS can be derived and appropriate unconditional calibration tests can be constructed. Moreover, the functional relationship between the two moments can be used to define a regression model in which the individual scores are regressed on their (suitably transformed) predictive variances leading to another procedure to test for miscalibration [36]. A theoretically well-founded way

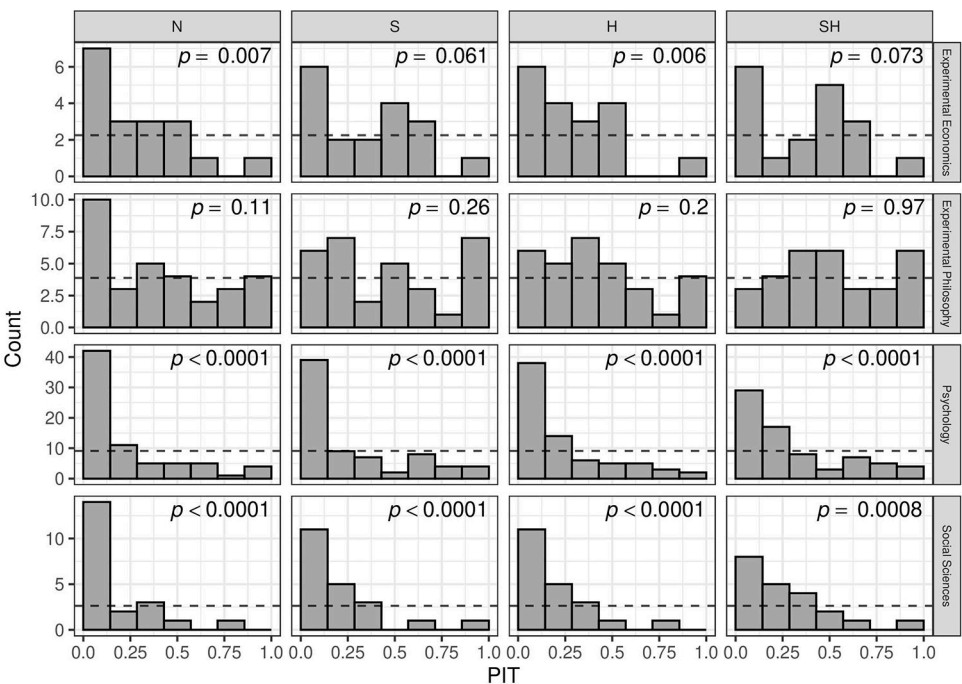

**Fig 8. Histograms of PIT values with *p*-values from Kolmogorov-Smirnov test for uniformity.** Dashed lines indicate number of counts within bins expected under uniformity. Forecasting methods are abbreviated by N for *naive*, S for *shrinkage*, H for *heterogeneity*, SH for *shrinkage and heterogeneity*.

to summarize the *p*-values of these tests is to use their harmonic mean $\overset{\circ}{p}$ [40–42], which is also shown in Table 1 (see S2 Appendix for non-summarized results).

Taken together, there is strong evidence for miscalibration of all forecasts in the psychology and social sciences projects. In the economics project, on the other hand, there is no evidence for miscalibration of the H and SH forecasts and weak evidence for miscalibration of the other forecasts. Finally, in the philosophy project there is strong evidence for miscalibration of the N and S forecasts and no evidence for miscalibration of the H and SH forecasts.

**PIT histograms.**    Fig 8 shows histograms of the PIT values of the four forecasting methods along with *p*-values from Kolmogorov-Smirnov tests for uniformity. In some of the histograms in the social sciences and economics projects there are bins with zero observations, however, these projects also have the smallest sample sizes. In the psychology and social sciences data sets, the N method shows extreme bumps in the lower range of the PIT values, while the histograms of the H, S, and SH methods look flatter, suggesting less miscalibration. In the economics data set, the PIT histograms also show bumps in the lower range, but to a much lower degree than in the psychology and social sciences data sets. Finally, in the philosophy data set the histograms look acceptable for all methods, suggesting no severe miscalibration.

## Forecasts of statistical significance

Since statistical significance of the replication study is one of the most commonly used criteria for replication success, in the following section the probability of significance under the investigated predictive distributions will be evaluated using methods suited for probabilistic forecasts of binary target variables. Moreover, in the social sciences and experimental economics projects these forecasts can also be compared to forecasts from the prediction markets. The

**Table 2. Observed and expected number of statistically significant replication studies along with *p*-value from $\chi^2$-goodness-of-fit test.**

| Project | Method | Observed | Expected | *p*-value |
|---|---|---|---|---|
| Experimental Economics | N | 11 | 15.0 | 0.012 |
| *n* = 18 | S | 11 | 13.6 | 0.16 |
| | H | 11 | 14.3 | 0.057 |
| | SH | 11 | 12.4 | 0.49 |
| | PM | 11 | 13.6 | 0.16 |
| Experimental Philosophy | N | 23 | 27.8 | 0.004 |
| *n* = 31 | S | 23 | 26.2 | 0.11 |
| | H | 23 | 26.5 | 0.076 |
| | SH | 23 | 24.1 | 0.65 |
| Psychology | N | 24 | 55.4 | < 0.0001 |
| *n* = 73 | S | 24 | 49.2 | < 0.0001 |
| | H | 24 | 53.5 | < 0.0001 |
| | SH | 24 | 45.9 | < 0.0001 |
| Social Sciences | N | 13 | 19.9 | < 0.0001 |
| *n* = 21 | S | 13 | 19.2 | < 0.0001 |
| | H | 13 | 18.9 | < 0.0001 |
| | SH | 13 | 17.6 | 0.006 |
| | PM | 13 | 13.3 | 0.89 |

Forecasting methods are abbreviated by N for *naive*, S for *shrinkage*, H for *heterogeneity*, SH for *shrinkage and heterogeneity*, PM for *prediction market*.

significance threshold $\alpha = 0.05$ for a two-sided *p*-value was used in all cases. Most of the evaluations were also conducted for smaller $\alpha$ thresholds and are reported in S2 Appendix.

**Expected number of statistically significant replication studies.** By summing up all probabilities for significance under each method within one project, the expected number of statistically significant replication outcomes is obtained and can be compared to the observed number, *e. g.* with a $\chi^2$-goodness-of-fit test (shown in Table 2). In general, the observed number of significant replication studies is smaller than the expected number for all methods in all data sets, yet the amount of overestimation differs between the methods. The overestimation is the smallest for the SH method and the largest for the N method across all data sets.

In the economics and philosophy projects there is no evidence of a difference between expected and observed under the S and SH method, whereas there is weak to moderate evidence of a difference for the N and H methods. In the social sciences and psychology projects, on the other hand, there is strong evidence for a difference between the expected and the observed number of significant replications for all methods, suggesting miscalibration of these forecasts. Furthermore, the expected numbers under the prediction market (PM) method in the economics and social sciences projects do not differ substantially from what was actually observed, providing no evidence for miscalibration of these forecasts.

**Brier scores.** In Table 3 the mean (normalized) Brier scores are shown for each combination of data set and forecasting method. The mean normalized Brier score [34] is shown because it enables the comparison of models across data sets in which the proportion of significant replications differs (*e. g.* in the psychology data set the proportion is much lower than in the others). It is computed by $BS_n = (BS_0 - BS)/BS_0$ where $BS_0$ is the baseline Brier score assuming that all replication studies are given an estimated probability of significance equal to the proportion of significant replications. Hence, $BS_n$ is positive if the predictive performance of the model is better than the baseline prediction.

**Table 3. Mean Brier score (BS), mean normalized Brier score (BS norm), and test-statistic with _p_-value from Spiegelhalter's _z_-test.**

| Project | Method | BS | BS norm | z | p-value |
|---|---|---|---|---|---|
| Experimental Economics | N | 0.271 | −0.139 | 2.5 | 0.013 |
| n = 18 | S | 0.226 | 0.048 | 1.1 | 0.25 |
| | H | 0.262 | −0.104 | 2.0 | 0.042 |
| | SH | 0.227 | 0.046 | 0.8 | 0.43 |
| | PM | 0.243 | −0.021 | 1.5 | 0.15 |
| Experimental Philosophy | N | 0.193 | −0.007 | 3.3 | 0.0009 |
| n = 31 | S | 0.173 | 0.097 | 2.1 | 0.039 |
| | H | 0.170 | 0.110 | 1.6 | 0.11 |
| | SH | 0.148 | 0.229 | 0.2 | 0.83 |
| Psychology | N | 0.394 | −0.784 | 10.0 | < 0.0001 |
| n = 73 | S | 0.335 | −0.518 | 7.8 | < 0.0001 |
| | H | 0.363 | −0.644 | 8.5 | < 0.0001 |
| | SH | 0.289 | −0.308 | 5.2 | < 0.0001 |
| Social Sciences | N | 0.346 | −0.468 | 7.2 | < 0.0001 |
| n = 21 | S | 0.324 | −0.374 | 5.5 | < 0.0001 |
| | H | 0.310 | −0.316 | 4.9 | < 0.0001 |
| | SH | 0.272 | −0.155 | 3.4 | 0.0006 |
| | PM | 0.114 | 0.519 | −2.0 | 0.044 |

Forecasting methods are abbreviated by N for _naive_, S for _shrinkage_, H for _heterogeneity_, SH for _shrinkage and heterogeneity_, PM for _prediction market_.

In the social sciences and psychology projects the predictive performance is poor for all statistical methods. Namely, all mean Brier scores are larger than 0.25, a score that can be obtained by simply using 0.5 as estimated probability every time and additionally all mean normalized Brier scores are negative. In the economics project, the S and SH methods achieve a positive mean normalized Brier score, while it is negative for the N and H methods. Finally, the forecasts in the philosophy project show the best performance, _i. e._ all methods except the N method achieve a positive mean normalized Brier score with the SH method showing the largest value. Moreover, the PM forecasts show a normalized Brier score of about zero in the economics projects, which is comparable to the statistical methods, whereas in the social sciences project, the performance is remarkably good, far better than all statistical forecasts in this project.

Table 3 also displays the results of Spiegelhalter's _z_-test. In the psychology and social sciences data sets the test provides strong evidence for miscalibration of all statistical forecasts, but only weak evidence for miscalibration of the PM forecasts in the social sciences data set. In the economics data set, on the other hand, there is no evidence for miscalibration of the S, SH and the PM forecasts and weak evidence for miscalibration of the N and H forecasts. Finally, in the philosophy data set there is moderate evidence for miscalibration of the N and S forecasts, but no evidence for miscalibration of the H and SH forecasts.

**Calibration slope.** Fig 9A shows the calibration slopes obtained by logistic regression of the outcome whether the replication achieved statistical significance on the logit transformed estimated probabilities. In all but the psychology project the confidence intervals are very wide due to the small sample size. Also note that it was not possible to obtain the calibration slope for the PM method in the social science project because of complete separation. In the psychology and social sciences projects, the slopes of all methods are considerably below the nominal value of one suggesting miscalibration. However, the H and SH methods show higher values

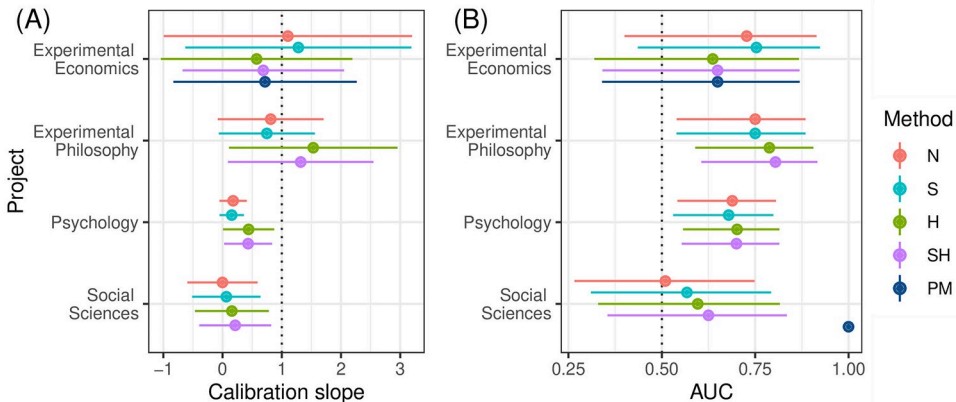

**Fig 9. Calibration slope and area under the curve (AUC) with 95% confidence interval.** Forecasting methods are abbreviated by N for *naive*, S for *shrinkage*, H for *heterogeneity*, SH for *shrinkage and heterogeneity*, PM for *prediction market*.

than the methods that do not take heterogeneity into account, indicating improvements in calibration. In the economics and philosophy projects, the slopes of all methods are closer to one and all confidence intervals include one, suggesting no miscalibration.

**Area under the curve.** Fig 9B shows the area under the curve (AUC) for each combination of data set and forecasting method. The 95% Wald type confidence intervals were computed on the logit scale and then backtransformed. Note that in the social sciences project for the PM forecasts, an AUC of one (without confidence interval) was obtained because the forecasts were able to completely separate non-significant and significant replications. The statistical forecasts in the social sciences project, on the other hand, show AUCs between 0.5 and 0.6 with wide confidence intervals, suggesting no discriminatory power. In the philosophy and psychology projects the H and SH methods show the highest AUCs. The former are around 0.8, while the latter are about 0.7, indicating reasonable discriminatory power of all forecasts. Finally, in the economics data set the N and S methods achieve the highest AUCs with values of around 0.75, but with very wide confidence intervals which all include 0.5.

## Sensitivity analysis of heterogeneity variance choice

For the H and SH methods the heterogeneity parameter $\tau$ was set to a value of 0.08 as described earlier. We performed a sensitivity analysis to investigate how much the results change when other values are selected. The change in predictive performance was investigated using the mean QS, mean LS, and mean CRPS, as they are good summary measures for calibration and sharpness of a predictive distribution. Furthermore, optimizing the mean score has been proposed as a general method of parameter estimation, which also includes maximum likelihood estimation (*i. e.* optimum score estimation based on the LS) [43] (Section 9).

Fig 10 shows the the mean scores for each project as a function of the heterogeneity $\tau$. In general, many of the mean score functions are rather flat, suggesting large uncertainty about the $\tau$ parameter. However, the chosen value of 0.08 seems plausible for the economics and philosophy projects, as it is close to the minima of all mean score functions. The values of $\tau$, which minimize the mean score functions in the social sciences and psychology projects, on the other hand, are substantially larger than 0.08. The SH model shows smaller mean scores than the H model over the entire range of $\tau$ in all but the philosophy data set, where both models show comparable mean scores. This suggests that evidence-based shrinkage leads to a better (or at

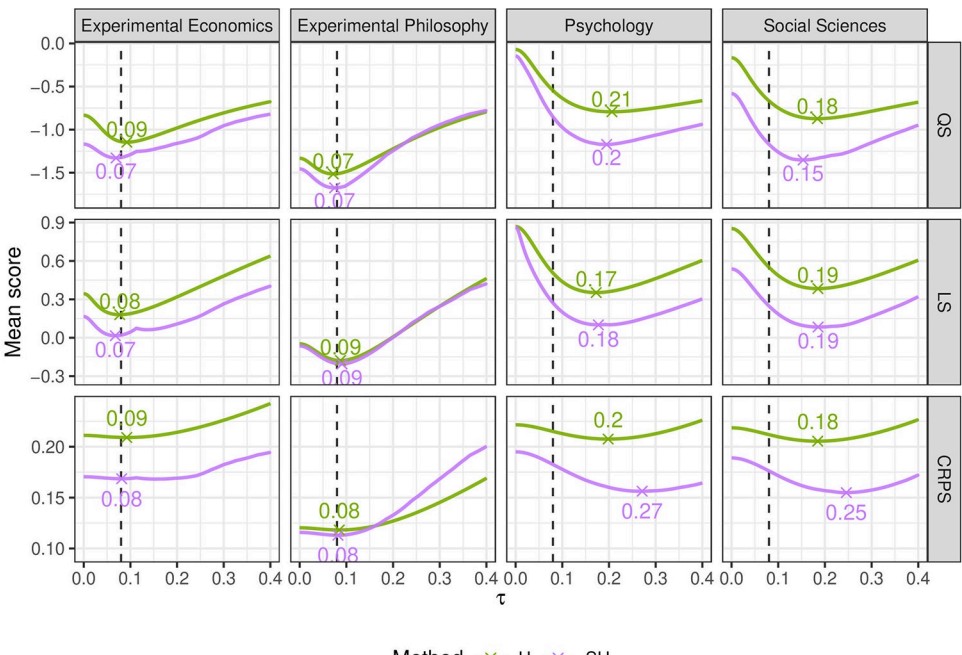

**Fig 10. Mean scores as a function of $\tau$ for each score type and project.** The dashed line indicates the chosen value of 0.08. Minima are indicated by a cross. Forecasting methods are abbreviated by H for *heterogeneity* and SH for *shrinkage and heterogeneity*.

least equal) predictive performance across all data sets and that the comparison of the methods is not severely influenced by the choice of $\tau$.

## Discussion

This paper addressed the question to what extent it is possible to predict the effect estimate of a replication study using the effect estimate from the original study and knowledge of the sample size in both studies. In all models we assumed that after a suitable transformation an effect can be modelled by a normally distributed random variable. Furthermore, we either assumed that in the original study the effect was estimated in an unbiased way (*naive model*), or we shrunk the effect towards zero based on the evidence in the original study (*shrinkage model*). In a Bayesian framework, the former arises when choosing a flat prior distribution for the effect, while the latter arises by choosing a zero-mean normal prior and estimating the prior variance by empirical Bayes. Finally, the models also differed in terms of whether between-study heterogeneity of the effects was taken into account or not, which was incorporated by a hierarchical model structure of the effect sizes.

Replication has been investigated from a predictive point of view before; Bayarri and Mayoral [18] used a similar hierarchical model but chose a full Bayesian approach with priors put also on the variance parameters. For the underlying effect, on the other hand, they chose a flat prior, which leads to a predictive distributions without shrinkage towards zero. Patil et al. [13] used a simpler model which was derived in a non-Bayesian framework, but corresponds to our naive model. This model was then used to obtain forecasts of replication effect estimates using the data set from the *Reproducibility Project: Psychology* [4] and also in the analyses of the *Experimental Economics Replication Project* [5] and the *Social Sciences Replication Project* [7]. In all of these analyses, however, apart from examining the coverage of the prediction intervals,

no systematic evaluation of the predictive distributions was conducted, even though there exist many well established methods for evaluating probabilistic forecasts. For this reason, we computed and evaluated the predictive distributions under the four different models for the three aforementioned data sets and additionally for the data from the *Experimental Philosophy Replicability Project* [8].

## Predictive evaluation

By taking into account between-study heterogeneity, evidence-based shrinkage, or both, calibration and sharpness have improved compared to the naive method. Forecasts obtained with the shrinkage and heterogeneity method usually showed a higher coverage of the prediction intervals, more uniformly distributed PIT values, substantially lower mean scores, and less or no evidence of miscalibration. The improvements have been larger in the social sciences and psychology and smaller in the economics and philosophy projects. However, in the psychology and social sciences projects, the tests still suggest some miscalibration, even for the heterogeneity and shrinkage model which performed the best, while there is less evidence for miscalibration in the philosophy and economics projects.

Furthermore, in the social sciences and economics data sets, the forecasts could be compared to forecasts from the non-statistical prediction market method which provides an estimate of the peer-beliefs about the probability of significance. In the economics data set, the shrinkage methods showed equal performance compared to the prediction market, while in the social sciences data set, the prediction market method performed better than any of the statistical methods.

It seems likely that in many of the investigated fields there is between-study heterogeneity present, as the models that take heterogeneity into account always performed the same or better than their counterparts which do not take heterogeneity into account. This is not surprising, as many of the replications used for example samples from different populations or different materials than those in the original studies [24]. Evidence-based shrinkage also improved predictive performance considerably in most cases, indicating that shrinkage is necessary to counteract regression to the mean. Moreover, this could suggest that the effect estimates from the original studies were to some degree inflated or even false positives, *e. g.* because of publication bias or the use of questionable research practices.

## Differences between replication projects

The predictive performance differed between the replication projects. There are several possible explanations for this phenomenon. The number of studies within a replication project could be one possible reason for the differences in the results of some of the evaluation methods, *e. g.* calibration tests. That is, the psychology project consists of many more study pairs, which leads to higher power to detect miscalibration in this project compared to the other projects.

Another explanation might be that differences in the study selection process of the replication projects lead to the observed differences. For instance, the original studies in the social sciences project were selected from the journals *Nature* and *Science*, which are known to mainly promote novel and exciting research, while in the philosophy, economics, and psychology projects they were selected from standard journals. Furthermore, if an original study contained several experiments, the rules to select the experiment to be replicated differed between the projects. In the psychology project, by default the last experiment was selected, whereas in the social sciences and philosophy projects by default the first experiment was selected. In the economics project, however, "the most central result" according to the judgement of the

replicators was selected by default. If on average researchers report more robust findings at the first position and more exploratory findings at the last position of a publication (or the other way around), this might have systematically influenced the outcome of the replication studies. Similarly, when replicators can decide for themselves which experiment they want to replicate, they might systematically choose experiments with more robust effects that are easier to replicate.

It may also be the case that the degree of inflation of original effect estimates varies between the different fields and that this leads to the observed differences. In particular, in the economics, social sciences, and psychology projects, the predictive performance was more substantially improved through evidence-based shrinkage than in the philosophy project, although the amount of shrinkage was roughly the same in all projects (see S2 Appendix for details). One possible explanation might be that experimental philosophy is less susceptible to publication bias, as it is a much younger field where there is high acceptance for negative or null results [8]. However, it may also be that in the early days of a field more obvious and more robust effects are investigated, which could explain the higher replicability of experimental philosophy findings.

## Conclusions

The attempt to forecast the results of replication studies brought new insights. Using a model of effect sizes which can take into account inflation of original study effect estimates and between-study heterogeneity, it was possible to predict the effect estimate of the replication study with good predictive performance in two of the four data sets. In the other two data sets, predictive performance could still be drastically improved compared to the previously used naive model [13], which assumes that the effect estimates of the original study are not inflated and that there is no between-study heterogeneity.

These results have various implications: First, state-of-the-art methods for assessing discrimination, calibration, and sharpness should be used to evaluate probabilistic forecasts of replication outcomes. This allows to make more precise statements about the quality of the forecasts compared to the ad-hoc methods which were used so far [5, 7, 12, 13, 15]. Second, researchers should be aware of the fact that original and replication effect estimates may show some degree of heterogeneity, although the study designs are as closely matched as possible. Finally, for the design of a new replication study, the developed model can also be used to determine the sample size required to obtain a significant replication result for a specified power. Our method provides a more principled approach compared to just shrinking the target effect size ad hoc by an arbitrary amount as was done in the planning of previous replication studies. Software for doing this as well as the four data sets are available in the R package `ReplicationSuccess` (https://r-forge.r-project.org/projects/replication/).

However, in the analysis of replication studies it may not be a good idea to reduce replication success solely to whether or not a replication study achieves statistical significance. One reason for this is that replication studies are often not sufficiently powered [44], so from a predictive point of view it is then not unlikely that non-significance will occur, even if the underlying effect is not zero. Another problem is that significance alone does not take into account effect size, *i. e.* significance can still be achieved by increasing the sample size of the replication study, even if there is substantial shrinkage of the replication estimate. We recommend instead to adopt more quantitative and probabilistic reasoning to assess replication success. Methods such as replication Bayes factors [45] or the sceptical $p$-value [46] are promising approaches to replace statistical significance as the main criterion for replication success.

Our results also offer interesting insights about the predictability of replication outcomes in four different fields. However, they should not be interpreted to mean that research from one field is more credible than research from another. There are many other factors which could explain the observed differences in predictive performance (see the discussion in the section "Differences between replication projects"). The complexity underlying any replication project is enormous, we should applaud all the researchers involved for investing their limited resources into these endeavours. There is an urgent need to develop new methods for the design and analysis of replication studies; these data sets will be particularly useful for these purposes.

The approach used in this paper also has some limitations: In all models, the simplifying assumption of normally distributed likelihood and prior has been made, which can be questionable for smaller sample sizes. Moreover, a pragmatic Bayesian approach was chosen, *i. e.* no prior was put on the heterogeneity variance $\tau^2$ and the variance hyperparameter of $\theta$ was specified with empirical Bayes. We recognize that a full Bayesian treatment, such as in the work of Bayarri and Mayoral [18], could reflect the uncertainty more accurately. However, our strategy leads to analytical tractability of the predictive distribution. This facilitates interpretability and allows to easily study limiting cases, which would be harder for a full Bayes approach where numerical or stochastic approximation methods are required. Moreover, it is well known that shrinkage is necessary for the prediction of new observations. The empirical Bayes shrinkage factor has proven to be optimal in very general settings [19, 30] and is for example also employed in clinical prediction models [38]. Furthermore, the data sets used all come from relatively similar fields of academic science. It would also be of interest to perform the same analysis on data from from the life sciences, as well as for non-academic research. Finally, only data from replication projects with "one-to-one" design were considered. It would also be interesting to conduct similar analyses for data from replication projects which use "many-to-one" replication designs, such as the "Many Labs" project [3, 6, 9], especially for the assessment of heterogeneity.

## Supporting information

**S1 Appendix. Data preprocessing.** Description and code for preprocessing of data. (PDF)

**S2 Appendix. Supplementary results.** Details and additional results. (PDF)

**S3 Appendix. Code.** Code to reproduce analyses, plots, and tables. (PDF)

## Acknowledgments

We thank Kelly Reeve for helpful comments on the draft of the manuscript.

## Author Contributions

**Conceptualization:** Samuel Pawel, Leonhard Held.

**Data curation:** Samuel Pawel.

**Formal analysis:** Samuel Pawel.

**Funding acquisition:** Leonhard Held.

**Investigation:** Samuel Pawel, Leonhard Held.

**Methodology:** Samuel Pawel, Leonhard Held.

**Project administration:** Samuel Pawel, Leonhard Held.

**Resources:** Leonhard Held.

**Software:** Samuel Pawel.

**Supervision:** Leonhard Held.

**Validation:** Samuel Pawel, Leonhard Held.

**Visualization:** Samuel Pawel.

**Writing – original draft:** Samuel Pawel.

**Writing – review & editing:** Samuel Pawel, Leonhard Held.

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
