## [Decision Letter · Decision Letter 0]

13 Jan 2020

PONE-D-19-30463

Probabilistic forecasting of replication studies

PLOS ONE

Dear Mr. Pawel,

Thank you for submitting your manuscript to PLOS ONE. After careful consideration, we feel that it has merit but does not fully meet PLOS ONE’s publication criteria as it currently stands. Therefore, we invite you to submit a revised version of the manuscript that addresses the points raised during the review process.

After hearing from the two reviewers and reading your manuscript I feel that the paper should undergo a major revision. Specifically, you should try to address all the issues raised by the reviewers. One of the reviewers suggested a minor review, the feedback from that report as well as the feedback of the other reviewer rather suggest a major revision.

We would appreciate receiving your revised manuscript by Feb 27 2020 11:59PM. To enhance the reproducibility of your results, we recommend that if applicable you deposit your laboratory protocols in protocols.io, where a protocol can be assigned its own identifier (DOI) such that it can be cited independently in the future. For instructions see: http://journals.plos.org/plosone/s/submission-guidelines#loc-laboratory-protocols

We look forward to receiving your revised manuscript.

Kind regards,

Petre Caraiani

Academic Editor

PLOS ONE

Journal Requirements:

'The preparation of this article was partially supported by the Swiss National Science Foundation (project number 189295 awarded to LH). The funders had no role in study design, data collection and analysis, decision to publish, or preparation of the manuscript.'

Additional Editor Comments (if provided):

Reviewers' comments:

Reviewer's Responses to Questions

**Comments to the Author**

1. Is the manuscript technically sound, and do the data support the conclusions?

Reviewer #1: Yes

Reviewer #2: Yes

2. Has the statistical analysis been performed appropriately and rigorously? 

Reviewer #1: Yes

Reviewer #2: Yes

3. Have the authors made all data underlying the findings in their manuscript fully available?

Reviewer #1: Yes

Reviewer #2: Yes

4. Is the manuscript presented in an intelligible fashion and written in standard English?

Reviewer #1: Yes

Reviewer #2: Yes

5. Review Comments to the Author

Reviewer #1: -----------------------------------------------------------------------------------------

Please see the attached pdf file for comments on the paper.

Reviewer #2: The authors use four replication projects each containing multiple studies and compare predictive models for the effect estimates based on methods known from the probabilistic forecasting literature. The predictive models considered here allow for heterogeneity in the effect sizes between the original study and the replication and for inflated effect estimates in the original study. While the idea is interesting, I am missing a discussion of what this novel approach might add and a more detailed comparison to existing work. This leaves me wondering what the takeaway lessons might be, besides reinforcing arguments already known from the literature.

I should note that my expertise lies in forecast evaluation, where no concerns arise. The technical details of the paper seem sound. I think, however, that the derivation of the "heterogeneity variance" of the effect estimates, could be explained and motivated in more detail (or changed altogether).

Main Comments

- Motivation of Approach

The authors state that they "will try to predict the effect estimates of the replication studies" with a novel prediction model allowing for heterogeneity and inflation of estimates and "compare them to the forecasts from the naive model" with "established evaluation methods from the statistical prediction literature". All those tasks are well executed. I wonder, however, what the paper contributes to the broader picture. In fact, for me the most interesting result is how the prediction market fares compared to the bayes predictions. Some statements in the abstract clearly are of general interest("estimates from the original studies were too optimistic...some degree of heterogeneity should be expected...statistical significance as the only criterion for replication success may be questionable"). However, those have been made before and I think the article is missing an argument why the predictive comparison is an adequate tool (compared to for example a full bayesian model) to add to those results. It strikes me as odd to construct Bayes predictions, compare different models, and then to make inference about model parameters(e.g., heterogeneity) based on predictive performance instead of estimating heterogeneity in a Bayes model. If, instead of inference about heterogeneity and inflation, the goal is the construction of a well-performing forecasting model, I would consider it more promising to have data driven models and compare them out-of-sample. Further, the authors could conclude what additional insights were gained from the more sophisticated tools (scores, PIT, etc.).

The additional arguments, explaining why the approach is interesting, could be accompanied by a more detailed comparison to the existing literature. I think, for example, that Bayarri and Mayoral (2002) also employ a hierarchical model that allows for heterogeneity of effect sizes. Mentioning such similarities and pointing out differences to the existing literature would certainly improve the paper.

- The section "Specification of the heterogeneity variance" should be improved

I have several issues with this section. First, the choice seems rather ad-hoc. I would have considered it more natural to formulate a prior over the heterogeneity parameter (or use other estimation methods), instead of assuming a fixed value. Your robustness analysis alleviates most of my concerns. So, your solution seems sensible enough, however, it took me quite some time to follow. What exactly is the "elicitation of opinion approach"? I don't think the mentioned reference gives a definition in Chapter 5.7.3, but rather applies it. I may be wrong. In any case, you could reconsider the explanation. I also think that the concept is normally meant to elicit priors from experts. I would advise to define your approach, explain it in detail, find suitable references, and discuss its implications in more detail. As part of this, let me point you to some details: (1) You write "[...]since this decision is only motivated theoretically", where I think "motivated heuristically" (or similarly) is more appropriate. (2) In line 216, "This suggests τ = 0.08 for δ(τ) being of the size of a medium effect." was confusing to me, as the argument before only discourages large effects, but not small effects. After assuming said medium effect size, you compute the respective τ. If this is indeed the case, I think this could be stated more explicitly. (3) It is unclear to me why the definition of effect sizes by Cohen should bear any weight in finding an appropriate variance parameter for heterogeneity. You could consider discussing this point. (4) $\\theta$ is introduced as (underlying) effect (l. 109), later called effect size (l. 131), which is now also used for $theta_k$ (l. 202). It would have been easier for me, if two different names would be used or if the symbols(theta,theta_k) would be used throughout. Finally, I should note that your results in the Section "Sensitivity analysis of heterogeneity variance choice" are insightful and a convincing argument for your choice.

Minor Comments

- As part of reconsidering the motivation and takeaways: The following sentence in the abstract puzzled me in the first reading. In hindsight, I know what you mean, but am still thinking this should be more precise: "...many of the estimates from the original studies were to optimistic, ...some degree of heterogeneity should be expected."

- Section numbering: I found the absence of section numbering confusing and myself often wondering if I am to embark on the next section or subsection now.

- Section labels "Continuous forecasts"/"Binary Forecasts". I think it may be more helpful to name the sections differently or mention more explicitly that one considers forecasts of the effect estimates and the other forecasts of the effects being significant. Maybe the confusion arose because it is actually the target variable which is continuous/binary and not the forecast. As part of this, you might reconsider terms like "binary predictive distributions" (l. 314), which are in fact probability predictions for a binary target/outcome variable with values on the unit interval.

- line 300: The KS test specifies the behavior under uniformity, the language "test for non-uniformity" should probably be reconsidered. I have similar doubts regarding the term "miscalibration tests", which actually test the hypothesis of a calibrated forecast. I think tests are best named in accordance with their hypothesis, not the alternative they potentially have power against.

- The PIT is considered a tool for assessing calibration. It is a bit odd to start with PIT-histograms, continue with scores, before testing calibration including the PIT uniformity test. Further, including the p-values of the uniformity test in the pit-histogram discussion (or plot) seems preferable to me.

- Calibration tests: If I understand the code correctly, you use regression based calibration tests. While this is mentioned before the results in line 236, it would be great to mention this again (with more detail) in the Section "Miscalibration tests" or in the Appendix S2 (which is referred to but unfortunately does not seem to contain any more details on this point).

- Personally, I would have appreciated captions with more details for the tables, eradicating the need to search the text for definitions.

- line 411 - 416: I think it is more consistent to state that you assume that the effect estimate was inflated(!). The forecasting model you use therefore is shrinking ("they shrunk the effect" sounds like the initial estimate was shrunk). Also, "can be achieved" seems an overstatement. Perhaps "can be modeled" is more appropriate.

- I would like to express my compliments for providing executable code and making the data sets available via an R-package.

6. PLOS authors have the option to publish the peer review history of their article (what does this mean?). If published, this will include your full peer review and any attached files.

Reviewer #1: No

Reviewer #2: No

---

## [Author Response · Author response to Decision Letter 0]

26 Feb 2020

All responses to the academic editor and the reviewers are contained within the file "ResponseToReviewers.pdf".

---

## [Decision Letter · Decision Letter 1]

24 Mar 2020

Probabilistic forecasting of replication studies

PONE-D-19-30463R1

Dear Dr. Pawel,

We are pleased to inform you that your manuscript has been judged scientifically suitable for publication and will be formally accepted for publication once it complies with all outstanding technical requirements.

With kind regards,

Petre Caraiani

Academic Editor

PLOS ONE

Additional Editor Comments (optional):

Reviewers' comments:

Reviewer's Responses to Questions

**Comments to the Author**

1. If the authors have adequately addressed your comments raised in a previous round of review and you feel that this manuscript is now acceptable for publication, you may indicate that here to bypass the “Comments to the Author” section, enter your conflict of interest statement in the “Confidential to Editor” section, and submit your "Accept" recommendation.

Reviewer #1: All comments have been addressed

Reviewer #2: (No Response)

2. Is the manuscript technically sound, and do the data support the conclusions?

Reviewer #1: Yes

Reviewer #2: Yes

3. Has the statistical analysis been performed appropriately and rigorously? 

Reviewer #1: Yes

Reviewer #2: Yes

4. Have the authors made all data underlying the findings in their manuscript fully available?

Reviewer #1: Yes

Reviewer #2: Yes

5. Is the manuscript presented in an intelligible fashion and written in standard English?

Reviewer #1: Yes

Reviewer #2: Yes

6. Review Comments to the Author

Reviewer #1: (No Response)

Reviewer #2: The revision improved the paper. I should note that I still have doubts on the scientific impact of the submitted manuscript, but all concerns on the methodology and discussion were addressed. To my best judgement, the study is appropriate for publication in PLOS One.

Minor comment:

- Maybe in the abstract the heterogeneity is more between effects and not effect estimates (as those are non-identical in literally every case)

"some degree of heterogeneity between original and replication effect estimates should be expected"

7. PLOS authors have the option to publish the peer review history of their article (what does this mean?). If published, this will include your full peer review and any attached files.

Reviewer #1: No

Reviewer #2: No

---

## [Editor Report · Acceptance letter]

26 Mar 2020

PONE-D-19-30463R1 

Probabilistic forecasting of replication studies 

Dear Dr. Pawel:

I am pleased to inform you that your manuscript has been deemed suitable for publication in PLOS ONE. Congratulations! Your manuscript is now with our production department. 

With kind regards,

on behalf of

Dr. Petre Caraiani 

Academic Editor

PLOS ONE